# Nodule Detection with Convolutional Neural Network Using Apache Spark and GPU Frameworks

**Nikitha Johnsirani Venkatesan [1], Dong Ryeol Shin [1] and Choon Sung Nam [2,*]**

1    Department of Electrical and Computer Engineering, Sungkyunkwan University, Suwon 10027, Korea; nikipraha.18@gmail.com (N.J.V.); drshin@skku.edu (D.R.S.)
2    Department of Software Convergence Engineering, Inha University, Incheon 15798, Korea
*    Correspondence: namgun99@gmail.com

**Abstract:** In the pharmaceutical field, early detection of lung nodules is indispensable for increasing patient survival. We can enhance the quality of the medical images by intensifying the radiation dose. High radiation dose provokes cancer, which forces experts to use limited radiation. Using abrupt radiation generates noise in CT scans. We propose an optimal Convolutional Neural Network model in which Gaussian noise is removed for better classification and increased training accuracy. Experimental demonstration on the LUNA16 dataset of size 160 GB shows that our proposed method exhibit superior results. Classification accuracy, specificity, sensitivity, Precision, Recall, F1 measurement, and area under the ROC curve (AUC) of the model performance are taken as evaluation metrics. We conducted a performance comparison of our proposed model on numerous platforms, like Apache Spark, GPU, and CPU, to depreciate the training time without compromising the accuracy percentage. Our results show that Apache Spark, integrated with a deep learning framework, is suitable for parallel training computation with high accuracy.

**Keywords:** lung nodule; Apache Spark; Convolutional Neural Networks; deep learning

## 1. Introduction

Lung cancer is one of the most prevalent cancers globally, with only 16% of cases being diagnosed at an immature stage. World cancer research fund global statistics states that 58% of lung cancer befalls in developing countries due to a lack of early detection. About 154,050 people die from lung cancer in the U.S. per year, as per the report of the American Cancer Society [1]. The survival percentage of lung cancer can be increased by detecting the nodules early in the patient. However, detecting lung nodule need the utmost attention of the radiologists. The nodule size is comparatively petite in the pre-cancer stage and is difficult to differentiate from other benign tissues. The CT scan and visual noise resolution make it troublesome to diagnose even for specialist radiologists. The existing systems, such as LungRAD, suffer from many false positives while detecting lung nodules. To overcome these challenges, an Artificial Intelligence (AI) model is essential in each hospital to detect the nodule. There are numerous approaches to detect lung nodules classified by predetermined models and features [2–5]. To this point, Artificial Intelligence (AI) [6] has been proven to be one of the most thriving creations in the medical industry. Machine learning [7] and deep learning [8] algorithms are used extensively for classification purposes. Convolutional neural network [9] is one of the successful neural network models for image processing. It demands an enormous amount of labeled training data, which is considered difficult to acquire in the medical field. However, even with radiologists' careful labeling, the model's accuracy might decrease due to visual noise. The image quality of CT scans influenced by radiation as the image quality is high with increased dosage [10], and vice versa. However, a high dosage of radiation has many side effects, which uplift the chances of cancer [11,12]. A reduced dose of radiation results in poor CT image quality with visual noise.

As aforementioned, erratic noise makes it tedious for the radiologist to distinguish between the nodules in the lung region [13]. For lower-dose examinations, the ability to determine relevant disease on noisier images will depend on various factors, including lesion intensity, lesion contrast associated with neighboring tissues, and image noise and sharpness. Applying data processing and image restoration approaches that minimize image noise while preserving spatial resolution makes it possible to increase the quality and diagnostic value of intrinsically noisy low-dose CT images. Therefore, noise removal plays an essential role as a pre-processing step in deep learning to enable the precise detection of lung nodules [14]. Deep learning is proven to be successful among researchers because of its ability to self-learn the feature values and provide an authentic outcome. NVIDIA's researchers performed trials on the ImageNet dataset to remove grains and noises without the need for observing the clean data [15]. The same technique can be applied to medical images where acquiring training data with ground truth is tedious. By proper model training, deep learning makes it achievable to enhance the image quality by removing noise and grains without training images. Deep learning can reconstruct the under-sampled CT images by training with the available data. It can extract more useful information from the unstructured data even without labeled knowledge by self-learning. This complexly integrated feature extraction and deep layers classification model easily overcomes the conventional machine learning algorithms [16].

Gaussian noise is an additive noise; removing it involves smoothing the distinct inside region of an image. These classical linear filters, such as the Gaussian filter, reduce noise efficiently but significantly blur the edges. A local measure of an image is to detect edges quantitatively and level them less than the rest of the picture. The contaminated pixels at the edges will be smoothed effectively using the multiple layers on the proposed Non-Gaussian Convolutional Neural Network (NG-CNN) architecture. Hence, all the regions of images are concentrated at numerous stages in terms of pixel by pixel. Thereby we attain qualified noise removed images. In our proposed Non-Gaussian Convolutional Neural Network (NG-CNN), we applied Gaussian noise removal as a foremost pre-processing step, which is later trained using the neural network's hidden layers. A significant part of noise removal is linear filtering and non-linear filtering. Non-Gaussian CNN's construction helps in embracing the progress in profound architecture, learning algorithms, and regularization methods into image denoising. The non-Gaussian CNN model can handle Gaussian denoising with the unknown noise level. Non-Gaussian eliminates the latent clean image in the hidden layers, which constructively remove noise. It increases flexibility and capacity for exploiting image characteristics, which ensures the CT scan image's fidelity, aiming for better classification accuracy. We trained the deep learning model with a labeled lung dataset to produce a minimum error rate. This paper is a continuation of our work [17].

This paper improved the algorithm and computed the difference to reduce training time and analyze the performance. We implemented the proposed algorithm in various platforms, Apache Spark, GPU, and CPU. General deep learning models in real-time are being trained by GPUs [18] and parallelized GPUs [19]. The training time of a complex model with many hidden layers would take even several days to converge. To reduce the converging time and to fine-tune the hyper-parameters, distributed training is the solution. Many researchers are trying to integrate deep learning and Spark to diminish the computation expense and to distribute the process [20]. Hence, we did the performance analysis by comparing the platforms, as mentioned earlier. Overall, our contributions in this paper are listed as follows:

- The proposed NG-CNN model shows a great accuracy percentage in detecting the lung nodule even when the nodule's size is less than 1.5 mm as we applied a noise filter as an extensive pre-processing technique. Measurement of correntropy is integrated with the autoencoder based deep neural network.
- We designed Apache Spark deep learning framework for our proposed NG-CNN model. Training time and performance are analyzed using detailed experimentation on various platforms, such as Apache Spark, GPU, and CPU.

- Large labeled lung CT scan dataset of size 150 GB has been used for evaluation, making it perfect for CNN training. The cumulative number of cases is more than 1600, with multiple slices per patient.
- Classification accuracy, sensitivity, specificity, and area under the ROC curve (AUC) of the model performance are compared with various combinations of CNN parameters and other deep neural networks.

The rest of the paper is organized as follows: Section 2 exhibits the literature survey related to our research work. Our proposed methodology is illustrated in detail in Section 3. Section 4 demonstrates the results of the experiments and discussion. Section 5 concludes the paper along with possible future work.

## 2. Related Works

In this section, the background works related to our methodology are discussed briefly. The first part of the section focus on papers that improve the accuracy percentage in the CNN model. The second part of the section scrutinize performance comparison between Apache Spark and GPU.

Krizhevsky et al. [21] trained deep convolutional neural network for classification of ImageNet dataset, which consists of 1.2 million HD images. The deep network consists of five convolutional layers, three fully connected layers, and one softmax layer. GPU was employed to train the dataset, and the authors proposed a novel regularization method called "dropout" to avoid overfitting. Kalinovsky et al. [22] implemented segmentation using deep CNN with experiments conducted in GPU Nvidia. The authors concentrated on the segmentation of the lung images to categorize similar lung image patches.

Experimentation was done in a small set of lung images with few hundreds of scans for both testing and training. Related works were done in References [23,24] from Cambridge University, where pixel-wise segmentation in lung scans was implemented. The deep model consists of an encoder and decoder layer and a pixel classification layer. In Reference [25], the authors constructed a new dataset by transforming the original images. The medical field lacks labeled training images. Changing the original dataset to create more data is convenient to minimize the error rate and uplift deep model accuracy. Experimentation was conducted with various datasets, such as malignant nodules, artificial geometric tumours, non-cancerous, and combined. The results show that the transformed dataset can better capture CT scans' features than the original dataset. Romero et al. [26], introduced the single layer and deep convolutional networks based on a prediction system for remote sensing data analysis. They launched the supervised deep convolutional network, which can better operate on multi and hyperspectral imagery fields. They concluded that the proposed method could not produce the optimal outcome with a high dimensional dataset with less labeled information.

The authors in Reference [27] concentrated on the dataset and separated the dataset based on the nodules' volume. They enlarged the large lung nodule dataset to train the model from 756 to more than 35,000. Random cropping also further expanded the dataset's count as the author believes a deep model requires a broad set of training data. Setio et al., in Reference [28], compared various deep learning algorithms and evaluated them by applying the LUNA16 dataset and reducing the confusion matrix's false positives. Algorithms, such as candidate detection, ISICAD, subsolidCAD, largeCAD, ETROCAD, and M5L, are used for comparison. The results showed that convolutional networks and the best combination of algorithms give a promising prediction for medical image analysis with a minimum false positive percentage.

Similarly, Refs. [29,30] use various deep models to compare and analyze the best suitable algorithm for detecting lung nodules. Two deep neural networks, such as convolutional and recurrent neural networks, are combined as nodular deep. The experiments are conducted with 1200 scans from the LIDC-IDRI dataset and are evaluated using metrics, such as sensitivity and specificity. Lo et al. [31] use sphere template double matching technique to search the possible nodule-like shapes in the lung image data and later used the

CNN algorithm for final classification. Anirudh et al. [32] use unsupervised segmentation to provide only a point label and used CNN for binary classification. Our work focus on pre-processing the CT lung dataset to remove the Gaussian noise, later final classification is done by our proposed model NG-CNN.

Gupta et al. [33] proposed a framework that combines a deep learning algorithm and Apache Spark to utilize Spark's in-memory computing power efficiently. Experiments ascertained that Spark is a better option for analyzing Big Data using a multilayer perceptron (MLP). DL4j (Deeplearning4j) applies deep neural networks on distributed Spark servers integrated with GPUs. Adam [34], the founder of DL4j, includes deep learning libraries with Spark and deploys the deep learning networks by parallelizing the dataset. Using distributed Spark, DL4j uses data parallelism and converges across the clusters of the machine. In experiments, distributing data across the clusters using the Resilient Distributed Data (RDD) property of Spark reduces converging time without compromising accuracy percentage. In Reference [35], Li et al. introduced heterospark, which is Spark, along with GPU accelerated to integrate the computation capability of both.

Comparing with GPU, Spark shows promising results due to the increased number of cores in a distributed setup. Spark has a significant benefit in terms of reducing the computation complexity of the neural networks model. Moritz et al., in Reference [36], exclusively developed a framework named SparkNet for deep models training in Apache Spark. Caffe library is used for deep models, and Spark RDDs are used for reading the dataset through a customized interface. They benchmarked the performance using the Imagenet dataset and altering the number of workers in the Spark server. The number of standard iterations and clusters is stated in the research paper to omit the communication overhead. From the preceding research papers, it is evident that deep learning accuracy entirely depends on the datasets' size. There is a better chance for a decent accuracy percentage in a deep model as we expand the dataset's size for training. Today, Big Data is flooding in zeta-bytes of data per minute from all over the world, such as social media, sensors, organizations, search queries, and the like. Massive databases are available to save zeta bytes of data, where machine learning and deep learning algorithms help to process the extensive amount of data to get valuable insights as the dataset gets larger, the training time for profound neural network advances in terms of weeks and even months. By distributing the data or model across the number of clusters reduces the computational cost and complexity. In the following section, we will discuss how we implemented our proposed model in Apache Spark, a vibrant ecosystem of deploying deep neural models.

## 3. Our Proposed Methodology

Our proposed method introduces the Gaussian noise removal technique as a foremost pre-processing step and integrating auto-encoder with a convolutional neural network. In a deep convolutional model, classification techniques give flexible lung nodule prediction, as there are multiple features present in the dataset. In this section, we also define the correntropy measure with equations. The autoencoder used in our proposed model and how our model deals with the Gaussian noise in the lung images are given in detail. Figure 1 provides the overall flow of our proposed lung nodule detection framework in five stages.

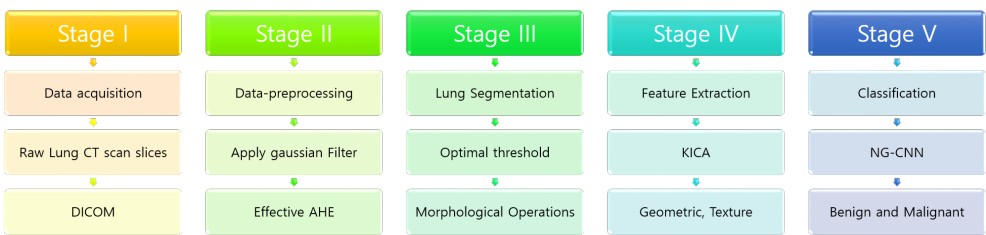

**Figure 1.** Overall flow of our proposed lung nodule detection.

### 3.1. Pre-Processing

Noise in which probability density function is equal to the standard pixel of images is Gaussian noise. It is obliged to predict the difference between the average pixel value and the Gaussian noised pixel value, which is more challenging due to their similarity. In this research method, we have done Gaussian noise exposure by using the correntropy measure. Correntropy is a kernel-based similarity measure that contains both the statistical and temporal structure of the underlying dataset. It mainly dispenses with the computation of accuracy, sensitivity, specificity. No comparison of correntropy is given in any existing methods, and our approach is specific as it deals with the parameters mentioned earlier. In our paper, the measurement of correntropy is integrated with the autoencoder based deep neural network. Autoencoders are based on unsupervised learning strategies, consisting of three components, encoder, code, and decoder. The function of an encoder is to compress the input and produce it to the system. The decoder reconstructs the image only using the code. Sometimes the model could over-fit the input data. To overcome this problem and learn a robust representation of the input data, we can manually add some noise called denoising auto-encoders. This integrated mechanism, namely Gaussian noise aware autoencoder, can ensure accurate and reliable detection of lung nodules even in the presence of Gaussian noise. Figure 2a,b shows the original lung CT scans and the pre-pro-cessed CT scans.

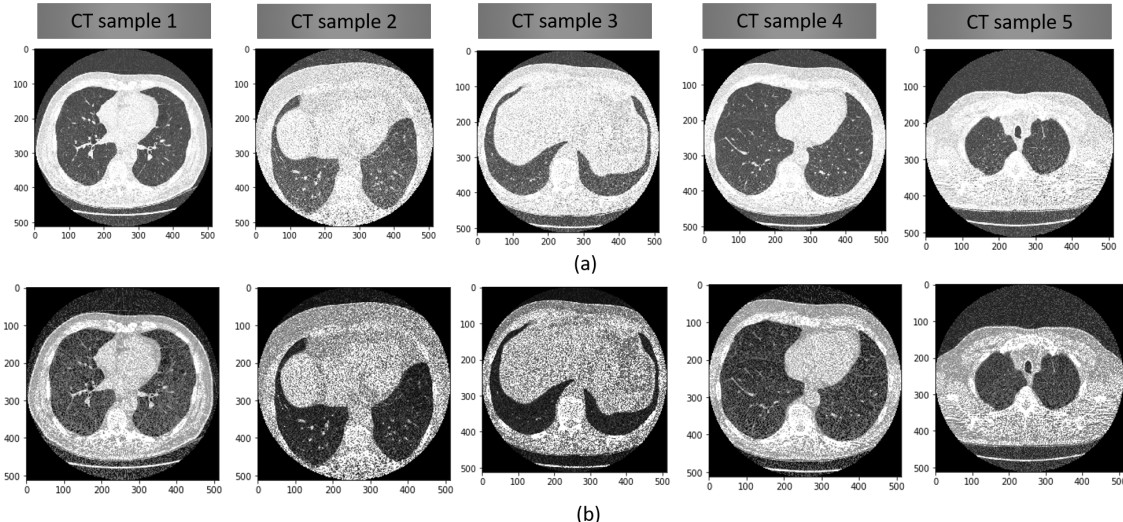

**Figure 2.** Results of pre-processed image and the original lung CT scans.

Our model uses an auto-encoder, where we trained the model to reduce the loss between the output from the decoder and the original noise-free image rather than the noisy CT image. We used the mean squared error to calculate the loss between the decoder's output and the encoder's input. For our lung dataset, $\{(a_i, b_i)\}_{i=1}^N$, where $a_i$ depicts the training dataset, and $b_i$ tells the ground truth labels. Loss over the dataset is given by the sum of all the losses calculated over each iteration.

$$L = \frac{1}{N} \sum_i L_i(f(a_i, W), b_i). \tag{1}$$

$L_i$ denotes the loss function for each batch of the dataset. $a_i$ represents the input dataset, $W$ denotes the weight value of the model, $b_i$ represents the bias value. Once the loss is calculated, we used back-propagation to update our weights throughout the network. After proper training, our model removes the maximum noise from CT lung images to attain better accuracy.

We adopted the correntropy measure to assess the similarity between the nearest pixels based. Correntropy measure between the two pixels, namely *A* and *B*, is rated as follows:

$$Correntropy_\sigma(A, B) = E[k_\sigma(A - B)], \tag{2}$$

where $E[.]$ represents mathematical expectation, $k_\sigma$ is Gaussian kernel function, and $\sigma$ describes kernel size. Correntropy measure is similar to renyi quadratic entropy [37] that is done to detect the similarities between the distributed data and ensure the exclusion of Gaussian noises present in the lung nodule dataset. The kernel function provided in equation two is estimated by the given Equation (3).

$$k_\sigma(.) = \frac{1}{\sqrt{2\pi\sigma}} exp\left(\frac{(.)^2}{2\sigma^2}\right). \tag{3}$$

The equation explicates that the correntropy measure is of limited value. The parameter $\sigma$ in the equation portrays the correlation adjustment factor. $\sum$ and high order moments are directly proportionate to each other, where the increased $\sigma$ value would also strengthen the higher-order moments. Thus, the equivalent distance would differ from 2 norms to zero norms if the spread between A and B increases. Henceforth, it distinguishes the irregularity and irrelevant data from the database precisely. The computation method of correlation with the lack of data about joint probability distribution function between *A* and *B* is given in Equation (4):

$$Corre\hat{n}tropy_\sigma(A, B) = \frac{1}{N} \sum_t^N k_\sigma(a_t - b_t). \tag{4}$$

The equations, as mentioned previously (2)–(4), predict the correntropy among the two single-pixel values. The calculation style of correntropy induced metric connecting two pixels vectors $P=(p_1,p_2,...,p_N)^T$ and $Q=(q_1,q_2,...,q_N)^T$ is described in the subsequent Equation (5):

$$CIM(P, Q) = \left(g(0) - \frac{1}{N} \sum_{t=1}^N g(e_i)\right)^{\frac{1}{2}} = \left(g(0) - \frac{1}{N} \sum_{t=1}^N g(p_i - q_i)\right)^{\frac{1}{2}}. \tag{5}$$

In the above Equation (5), $e_i$ depicts the error value which is calculated by Equation (6), and $g(x)$ is Gaussian kernel which is calculated by using (7).

$$e_i = p_i - q_i, \tag{6}$$

$$g(x)^\Delta = exp\left(-\frac{x^2}{2\sigma^2}\right). \tag{7}$$

In the above equation, $\sigma$ determines the width of the kernel. The square of the Gaussian probability density function $\sigma^2$ is variance where $\sigma > 0$. The maximum correntropy values of error $e_i$ are calculated as per the following (8):

$$max\frac{1}{N} \sum_{t=1}^N g(e_i). \tag{8}$$

The purpose of auto-encoders is to acquire the features with a minimum reconstruction cost function. The proposed NG-CNN with autoencoder is a deep model with three hidden layers: encoder and decoder. The layer's network structure in the pre-processing model consists of one input layer with d inputs, one hidden layer, one reconstruction layer, and one activation function. Gaussian noise removal is the primary interest to eliminate the visual noise from the input lung nodule dataset. By completing the noise removal, our model ensures an accurate learning rate. The encoder from Gaussian noise removal will assign the input vector $a \in R^d$ to the hidden layer, and we generate the intrinsic activity, which is portrayed as $b \in R^h$. The inherent activity value b will then shift by a decoder to

the output layer, where the input remodeling is performed. The output derived from the reconstruction process in the output layer is $c \in R^d$. The mathematical prognosis for values $b$ and $c$ is produced by:

$$b = f(W_b a + q_b), \tag{9}$$

$$c = f(W_c b + q_c). \tag{10}$$

$W_b$ is the input supplied to the hidden layer weights, $W_c$ is hidden to output layer weights, $q_b$ is the bias of hidden layer, $q_c$ is the bias of output layer, and, finally, $f(.)$ is the activation function.

$$\sigma(x) = \frac{1}{1 + e^{-x}}. \tag{11}$$

In Equation (12), the weights of noise removal autoencoder is calculated. The parameters are given as $\theta = \{W, q_b, q_c\}$, which reconstruct the input data values from the output data values with a standardised rebuilding cost function. The main objective of the proposed research model is:

$$W_b = E_c^t = W. \tag{12}$$

The reconstruction cost of Gaussian noise removal autoencoder is equated by (13) with the concern of mean square error and cross-entropy values among the input vector and output vector values. The loss function is mean square error, and the noisy CT image is loaded into the model.

$$J_{cost}(\theta) = L(a, c) + \lambda ||J_f(a)||_F^2. \tag{13}$$

In the above equation, $\lambda$ is positive hyper-parameter that is used to control the regularization of the deep model parameter values, and $J_{cost}(\theta)$ is the correntropy cost function. The reconstruction cost function is defined as:

$$L(a, c) = \frac{1}{m} \sum_{t=1}^{m} \sum_{k=1}^{n} k_\sigma (a_t k - c_t k). \tag{14}$$

In Equation (14), m is the number of input training samples of lung nodule scan images, and n is the training samples' length. To defend the robustness, we apply Jacobian norm mapping $J_f(a)$. It is a non-linear mapping value of encoding function f. This is adopted to map the hidden representation which is illustrated as $h = f(x) \in R^d h$. The summation of extracted features from the lung nodule CT images is calculated as:

$$\begin{aligned} ||J_f(a)||_F^2 &= \sum_{t=1}^{d_h} \sum_{j=1}^{d_x} \left( \frac{\vartheta h_i}{\vartheta x_j} \right)^2 \\ &= \sum_{t=1}^{d_h} \sum_{j=1}^{d_x} \left( h_1 (1 - h_1) \times w_{ij} \right)^2 . \\ &= \sum_{t=1}^{d_h} d_h \left( h_i (1 - h_i) \right)^2 \times \sum_{j=1}^{d_x} W_{ij}^2 \end{aligned} \tag{15}$$

Based on the above-computed norm values, the reconstruction cost of the proposed feature learning and lung nodule detection has a maximum accuracy percentage. The computation complexity of the proposed deep NG-CNN model is $O(d_x \times d_h)$. The pseudo-code of our proposed NG-CNN algorithm for pre-processing step is given in Algorithm 1.

After extracting the features using autoencoder for minimum cost reconstruction, we will obtain efficient features. Following this, the extracted features are given to the classifier, i.e., the proposed NG-CNN architecture; the number of layers in the classifier parts includes three convolutional layers, 3 rectified linear units and three max-pooling layers with fully connected layer and a softmax layer. Data will be partitioned for testing and training. Due to adequate validation, the model will remove the noise. In theory,

the mathematical expressions are proved by implementing our proposed deep model in the various platforms.

---

**Algorithm 1** Pseudocode of pre-processing step in NG-CNN algorithm.

---

**INPUT:** Set of lung CT images $X = (x_1, x_2, ..., x_N)$ with ground truth labels $Y = (y_1, y_2, ..., y_N)$
**OUTPUT:** Binary classes 0 or 1 for cancerous and non-cancerous nodules.

1: Read the original images and Initialize the variables and parameters, $W_b$ <- hidden layer weights, $W_c$ <- output layer weights, $q_h$, $q_c$ <- bias of hidden layer and output layer
2: Calculate the $f(.)$ <- activation function, with respect to the noisy features.
3: Evaluate the kernel function based on positive and bounded value.
4: Calculate CIM between two variables $P = (p_1, p_2, ..., p_N)^T$ and $Q = (q_1, q_2, ..., q_N)^T$
5: Calculate Error non-linearity that is asymptotically uncorrelated at steady state $e_i = p_i - q_i$
6: Calculate cost function as $J_{cost}(\theta) = L(a, c) + \lambda ||J_f(a)||_F^2$
7: Compute the maximum correntropy measure value error before giving the features to the auto-encoder.

---

### 3.2. Segmentation and Candidate Nodules

After pre-processing, the lung CT scans are segmented to exclude the irrelevant background. We applied the region seed growing method for segmenting the ROI from the CT scans. The greyscale image is transformed into a binary image using the threshold of $-340$ HU. We employed morphological operations, like erosion and dilation. The erosion is applied with a disk radius of 2 mm to eliminate the tissues in the walls and parenchyma. Dilation is practised with a disk radius of 10 mm to retain the nodules that occur in the cavity. Finally, the original image is superimposed with the binary image to attain the segmented region of interest. Figure 3 shows the segmented results of the sample lung slices (a) shows the anatomy scan of a human body, (b) shows the pre-processed image, (c) shows the binary image based on the threshold. (d,e) shows the results of the CT after erosion and dilution respectively. Finally, 3(e) shows the superimposed image.

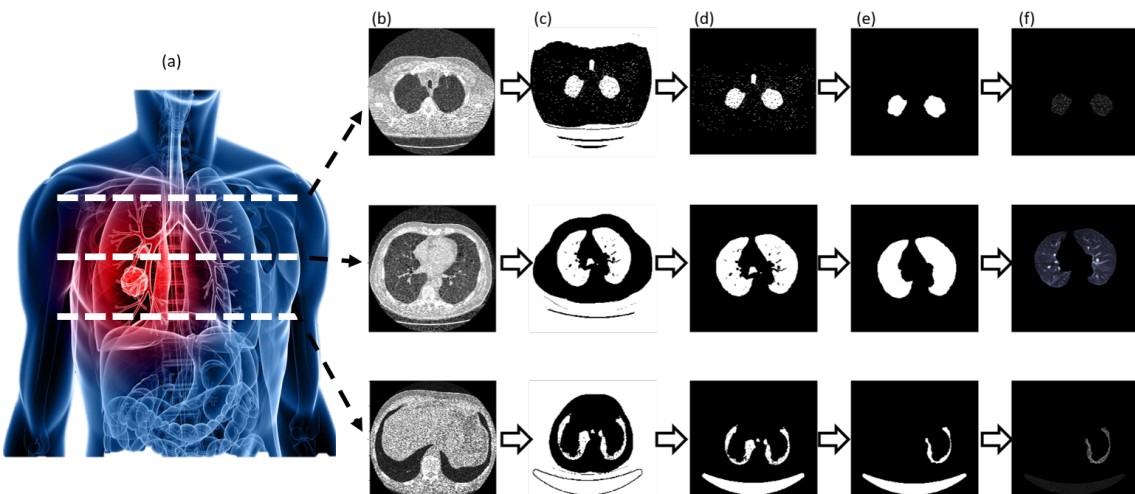

**Figure 3.** Sample results of segmented results of a top slice, middle slice, and bottom slice of a lung image of a patient.

### 3.3. Feature Extraction

After segmentation, the features are extracted based on geometric, intensity, and contrast. We extracted more than 240 features during the initial phase for all the potential nodule candidates. The 240 [38] features were shortlisted to 14 classes based on linear independence and ROC curve criterion. Features classes are based on geometric shapes, texture, intensity, gradient, spatial context, blobness, Eigen values, border, Radiomic features, kurtosis, skewness, graylevel, contrast, and Hessian. The intensity features are extracted based on the histogram values of the CT scan. The geometric features contains information about size and shape of the nodule. Features, such as radius, area, perimeter, compactness, roundness, and smoothness, are included as candidate nodular region. The texture features provide information on the variation of intensity by analyzing the characteristics, such as roughness and regularity. The selected candidate nodules are shown in Figure 4, which contains many false positives and false negatives. A deep model is applied in the section to reduce false positives.

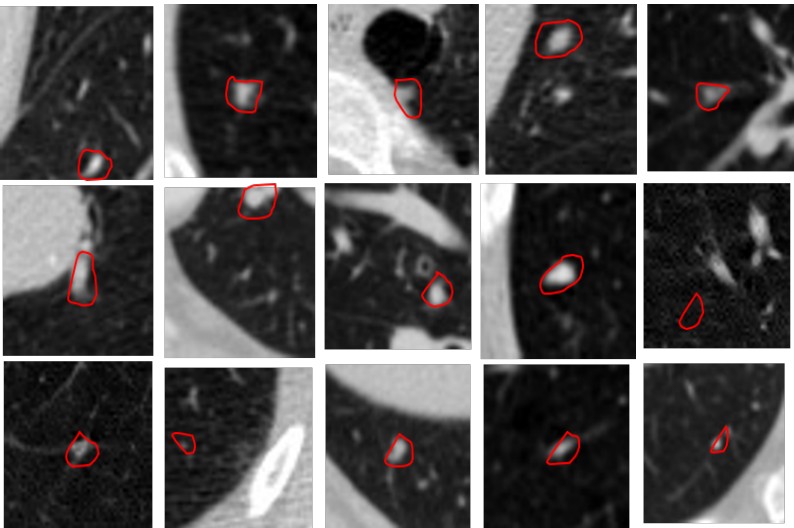

**Figure 4.** Selected nodule candidates from the features of the lung CT scans.

### 3.4. Training the Neural Net

This subsection details the working of CNN and the number of layers of our neural network. A CNN typically has three layers: a convolutional layer, a pooling layer, and a fully connected layer. The convolutional layer implements a dot product between two matrices, where one matrix is the set of learnable parameters otherwise known as a kernel, and the other matrix is the restricted part of the receptive field. During the forward pass, the kernel slides over the height and width of the image-producing the image representation of that sensory region. This produces a two-dimensional model of the image known as an activation map that gives the kernel's response at each scan's spatial position. The sliding size of the kernel is called a stride. The pooling layer replaces the network's output at specific locations by deriving a summary statistic of the nearby results. This diminishes the spatial size of the representation, which decreases the required amount of computation and weights. The pooling operation is processed on every slice of the model individually. Neurons in this layer have full connectivity with all neurons in the preceding and succeeding layer, as seen in regular FCNN. The FC layer helps to map the representation between the input and the output.

Figure 5 depicts our proposed model's architecture, defining the number of convolutional layers, ReLu layers, max-pooling, and fully connected layers. Our architecture has three convolutional layers+ ReLu layers and three max-pooling layers with one fully connected layer and a softmax layer. The first layer, the convolutional layer, maps our lung image dataset with multi-dimensional filters, which give us the first layer output. This in-

termediate output is fed as an input to the next layer. There are many kinds of pooling layers: max pooling, average pooling, sum pooling, and so forth. We chose max-pooling as it gives the maximum value in all the patches from the previous layer. Rectified Linear Unit is deployed as an activation layer for our proposed NG-CNN, which solves the adverse value problems and avoids computation expense f(x) = max(x,0).

For our lung nodule dataset, the output can have many classification stages, such as: (i) the patient has no nodule if the nodule size is less than 1.5 mm, (ii) the patient has a treatable nodule if the size of the nodule falls within 1.5 mm to 3 mm, and (iii) the patient has severe nodule if the nodule size is more significant than 3mm. However, there are still many stages in nodules, as most of them are treatable. If the patient has a nodule of size more than 1.25 mm thickness in our proposed methodology, then the algorithm's output will be positive, stating immediate attention. The outcome will be negative if the nodule size is less than 1.25 mm, resulting in binary classifications. Finally, the softmax layer is applied to deal with the classification process, and the function is used to categorize the probability distribution. Hence, the values are activated from 0 to 1, and the category with the highest probability is considered output. The experimentation results are provided in Section 4 exhaustively.

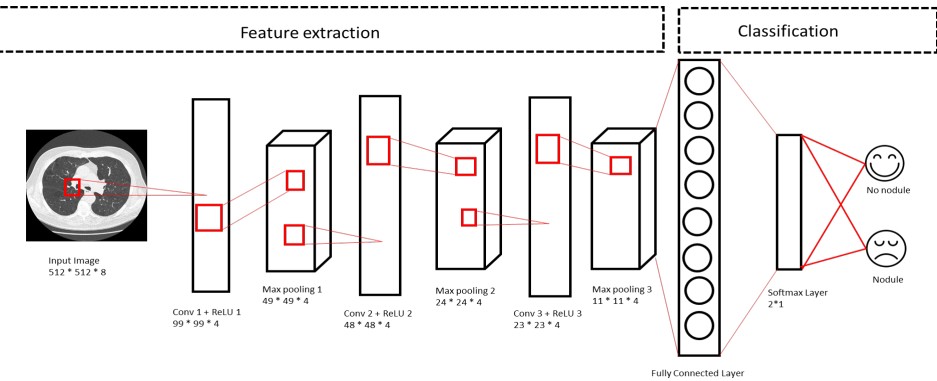

**Figure 5.** Selected nodule candidates from the features of the lung CT scans.

## 4. Results & Discussion

### 4.1. Dataset

We examined our deep model in experiments using an openly available LUNA16 dataset of volume 160 GB. According to the scan, the total number of patients is more than 1600, with variable slices per patient. The dataset contains low dose CT images in DICOM (Digital Imaging and Communications in Medicine) format for capturing the metadata of patients and other medical imagery. Medical professionals commonly use DICOM. The header file is included in the dataset with essential information, such as patient ID and the slice's thickness. The LUNA16 dataset has diverse image quality as it is taken from several hospitals and multiple machines. The labels for the training dataset were confirmed by pathology diagnosis. Four expert radiologists diagnosed every patient, and the ground truth label file is annotated. The patient with a nodule size greater than 1.25 mm is diagnosed as malignant, and those who have less than 1.25 mm are labeled as benign. All the patients have an individual folder containing multiple DICOM files with their id. The training and testing set is annotated along with ground truth labels of patients in CSV format with nearly 70% of patients without cancer, and the rest 30% diagnosed otherwise. The annotation file contains 1186 nodules.

### 4.2. Experimentation

In our paper, we have done experimentation to solve two problems. (1) Early Lung nodule detection (2) Minimizes the training time and speeds up the model's convergence. The initial segment explains the ROC and error rate of our proposed model CNN compared to other prevailing neural networks. The second part talks about distributed computing

using Apache Spark to converge the algorithm with minimal training time. This gives a swift opportunity to tune the best hyperparameters for our model. We compared the results of performance analysis on three different platforms CPU, GPU and Apache Spark.

We trained and evaluated our CNN model using four-fold cross-validation, and separate untouched data is used for testing. We randomly split the training data into four subsets of equal size for cross-validation. Hyperparameters are fine-tuned from the validation results, such as the number of filters K, their spatial extent F, the stride S, and the amount of zero padding P. TensorFlow is used as a deep learning library used in our CNN model with Nvidia GeForce GTX 750 GPU setup with 16 GB RAM. To deal with multiple format, DICOM files libraries, such as pydicom, matplotlib, and NumPy, are used. The CT image's pixel size is 512 × 512 with a depth of 195, which is very large and cannot be fed directly into our model because of the high computational cost. Pre-processing of the dataset is done to resize the data and control uniformity among all the patient's scans. The slices provided in the dataset's scans are not uniform, even though the image is of similar size. Our proposed CNN model used three convolutional layers. ReLU is used as an activation layer in our work [39], three max-pooling layers, along with pre-processing noise removal layer.

Furthermore, we handled one fully connected layer and a softmax layer as an activation function. The pixel size is resized to 99 × 99 without padding to minimize the computing complexity. In the max-pooling layer, the lung scan's pixel size is 49 × 49 as the upsampling layer is used to reduce the pixel values. This reduction is entirely based on the trained samples from which the region of interest is detected. We set the batch size to 160 with a complete training image set of 25,600. 15 × 15 convolution filter and 2 × 2 for pooling operation are applied in our algorithm, and the stride is chosen randomly with a pixel size of 7 × 7. In the second denoising layer, the pixel size is 48 × 48 and 24 × 24 during max pooling for the second time. Finally, in the third convolutional layer, the lung image is 23 × 23 and max pooled to 11 × 11. We apply noise removal as a pre-processing step where the pixel size is reduced to 23 × 23. Furthermore, after each convolutional layer, we use noise removal in every iteration for visual noise elimination.

The softmax layer of our methodology is 2 × 1. Our dataset size is 160 GB in total, and for training, we used 1595 patients. The quantity of slices for each patient varies according to the scans taken at the hospital. The slices per patient are put into a particular number of chunks. The input batch size is 160, and 40 iterations per epoch are used to train the samples with a learning rate of $10^{-2}$ and gradually decayed over the process.

The iterations are set to 10,000 with 250 epochs and four-fold cross-validation. We evaluated our accuracy percentage based on the number of epochs. Similarly, for the ROC curve, training accuracy, validation accuracy, training loss, and validation loss number of epochs are taken as x-axis for better percentage comparison. The performance is improved by choosing the best hyperparameters, such as the number of nodes, size, and the number of the filter from cross-validation results. Classification diagnostic accuracy, sensitivity, specificity, and area under the receiver operator characteristic curve (AUC) of the model performance is compared with various neural network models, such as Convolutional Auto Encoder Deep Learning Framework (CAE-DLF), Convolutional Neural Network (CNN), and Deep Belief Network (DBN). The accuracy of the diagnostic exam in the medical field is determined by sensitivity and specificity. The former shows the number of patients with the disease, and the following measures the false positive rate, which is misdiagnosis. The classification is analyzed using the receiver operator characteristic curve and area under the receiver operator character curve [40].

$$Sensitivity = \frac{TP}{TP + FN} * 100\%, \tag{16}$$

$$Specificity = \frac{TN}{TN + FP} * 100\%, \tag{17}$$

$$Accuracy = \frac{TP + TN}{TP + TN + FP + FN} * 100\%. \tag{18}$$

As mentioned earlier, we used four-fold cross-validation to choose optimal hyperparameters for our NG-CNN model. We ensured that training and testing datasets are from different patients to get an actual accuracy percentage. CNN, NG-CNN, CAE-DLF, and DBN are trained and tested with the same dataset. Figure 6 shows the validation and training loss for our proposed NG-CNN model in the top-left graph. The validation loss is 0.7256, which is higher than the training loss of 0.5148. The values are achieved after applying dropout to reach the minimum difference between training and validation loss. The bottom figures, Figure 6c,d, depict our proposed model's test accuracy and ROC curve with other deep models.

The test accuracy for our model is 0.9452. This accuracy percentage is more significant than the convolutional neural network without any noise removal step. The test accuracy percentages for our proposed model NG-CNN and other comparative methods are given in Table 1. Table 2 shows a comparison of the precision of NG-CNN with various neural network algorithms. Thus, it proves that medical images with visual noise reduction aim at better feature learning and accurate prediction of the lung cancer nodules. We compared the ROC curve for three deep neural models with our proposed model in Figure 6d. The x-axis indicates 1-Specificity, and the Y-axis indicates sensitivity. The figure represents the average graph of ROC achieved by four-fold cross-validation. Our proposed method, NG-CNN, has an AUC of 0.896, which performs better than other neural network models. Since higher AUC indicates maximum performance on average, NG-CNN gives less validation loss. Table 3 compares the sensitivity and false-positive rate with other models.

**Table 1.** Comparison of classification test accuracy of NG-CNN with existing methods.

| ep | NG-CNN | CAE-DLF | DBN | CNN |
|---|---|---|---|---|
| 50 | 0.5879 | 0.5124 | 0.5469 | 0.6789 |
| 100 | 0.78632 | 0.6645 | 0.6214 | 0.7356 |
| 150 | 0.8021 | 0.6987 | 0.6987 | 0.78541 |
| 200 | 0.8432 | 0.7421 | 0.7548 | 0.8632 |
| 250 | 0.9584 | 0.8952 | 0.79215 | 0.9452 |

**Table 2.** Comparison of Precision of NG-CNN with existing methods.

| ep | NG-CNN | CAE-DLF | DBN | CNN |
|---|---|---|---|---|
| 50 | 93.3150 | 90.3701 | 88.0467 | 95.7059 |
| 100 | 92.1970 | 89.1087 | 96.1765 | 96.5224 |
| 150 | 95.7167 | 95.9751 | 87.0868 | 99.7587 |
| 200 | 95.0442 | 91.4021 | 87.1534 | 98.6222 |
| 250 | 97.4947 | 89.9001 | 93.8550 | 97.5653 |

**Table 3.** Performance summary of NG-CNN with existing methods.

| Methods | Sensitivity | Specificity |
|---|---|---|
| NG-CNN | 91.3974 | 90.325 |
| CAE-DLF | 84.8537 | 79.215 |
| DBN | 88.6088 | 75.215 |
| CNN | 89.2759 | 89.214 |
| [41] | 71.0 | 60.0 |
| [42] | 80.0 | 75.0 |
| [43] | 90.0 | 88.0 |
| [44] | 76.0 | 73.0 |

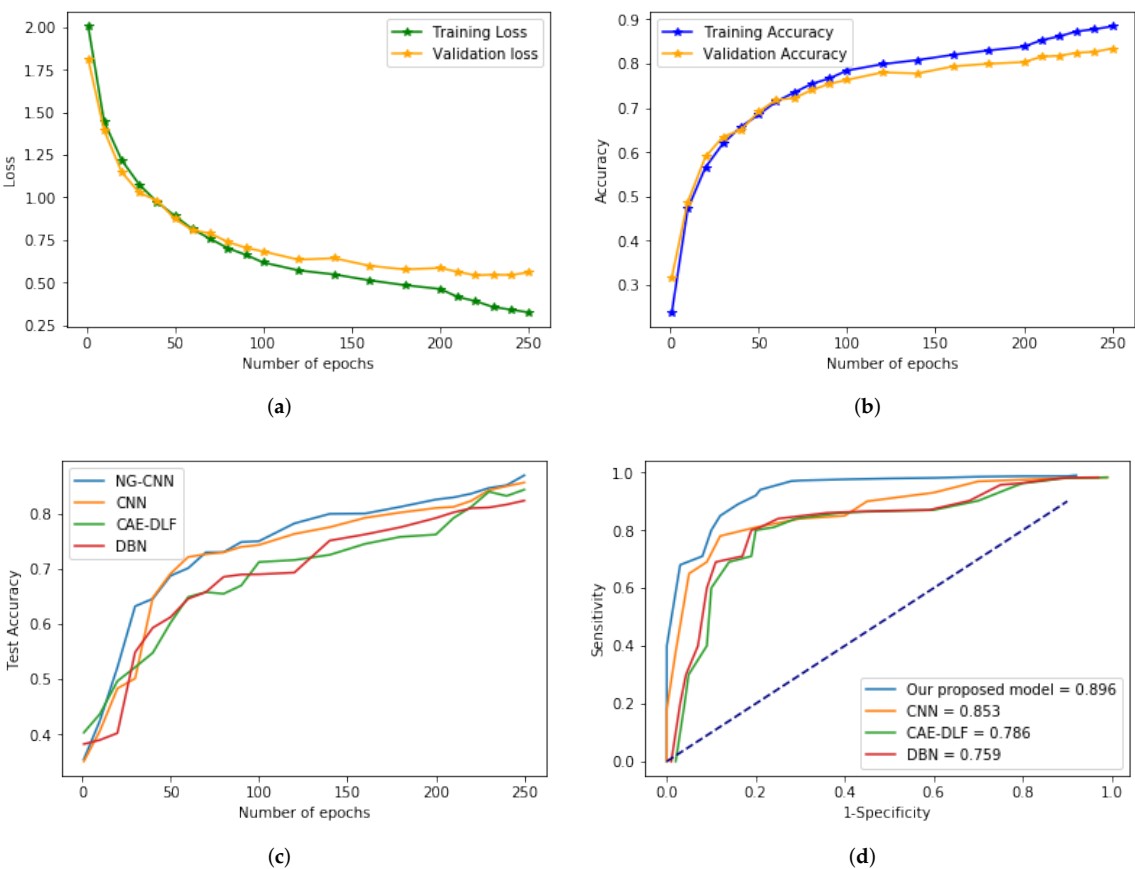

**Figure 6.** Comparison of the proposed method (NGN-ACDNN) Non-Gaussian Noise aware Autoencoder Convolutional Deep neural network with other deep models (CAE-DLF) Convolutional Autoencoder Deep Learning Framework, (CNN) Convolutional neural Network, (DBN) Deep Belief Network: (**a**) describes the Training loss and Validation loss for our proposed CNN model; (**b**) describes the Training accuracy and Validation accuracy for our proposed CNN model; (**c**) describes the Comparison of test accuracy of our proposed model with other deep neural network models; (**d**) describes the ROC curve comparison of our model with others.

### 4.3. Performance Analysis

For uniform platform comparison, we carried out the experiments in Ubuntu 14.04 Linux OS, Apache Spark v2.3.0, and Python v3. The size of the dataset is extensively large, which makes it challenging to work. The lung's raw image size is 512 × 512, roughly 40 million pixels, which take up 1 GB of system memory to load the data. A significant hindrance is to load the patient's data for every training step and iteration. Loading the data takes up even more time than training the same. Detailed experimentation was done by comparing it with Apache Spark, GPU, and CPU. The accuracy of the deep learning model is directly proportional to the size of the training data. When astronomical data is fed, the network architecture shows impressive performance. Training a massive amount of data on a single machine is computationally challenging. If the dataset cannot fit into a single machine's memory, training the neural network in distributed clusters will save the training time and memory. Hence, to analyze the training time and efficiency, comparative experiments on data parallelism is conducted in this section.

Spark operations that sort, group, or join data by value need to transfer data between partitions when constructing a new DataFrame from an existing one between stages, in a process called a shuffle. With a physical plan for CPUs, the DataFrame data is transformed into RDD row format and usually processed one row at a time. Spark supports columnar batch, but in Spark 2.x, only the Vectorized Parquet and ORC readers use it. Figure 7 shows that data is grouped by value and exchanged between partitions

(white rectangles) when creating a new DataFrame (blue rectangles) from an existing one between stages. Spark deep learning usually parallelize the dataset instead of the model itself. Figure 8 shows the general architecture of the distributed spark cluster. The data is distributed among the Spark workers. The gradients are updated via the parameter server in each iteration. Since the lung dataset we used is comparatively medium-sized, and one server was enough to process it, we did not parallelize the workers' deep learning model. Lung image data is parallelized among the workers by parameter averaging. Training the model in a cluster consists of the following steps.

- Parameters, such as weight, and biases are randomly initialized based on NG-CNN architecture.
- First copy of the parameters is distributed to all the Spark worker nodes.
- The lung image dataset is divided upon the worker nodes, and they train their subset of data.
- Update the calculated parameters after a certain number of iterations from the server node.
- Repeat from step 2 till the training converges.

The equation for parameter averaging in case of 2 worker nodes is given below.

$$W_{i+1} = \frac{1}{2} \sum_{j=1}^{2} W_{i+1,j}, \tag{19}$$

where $W$ represents the parameters, and i indicates the current values and $j$ the updated values. Similarly, in case of 4 worker nodes and one server, the parameter averaging would take place as follows:

$$W_{i+1} = \frac{1}{4} \sum_{j=1}^{4} W_{i+1,j}. \tag{20}$$

The worker nodes hold a copy of our neural network model, and the lung images are distributed among the nodes. The hyper-parameters such as the learning rate, number of epochs and iterations, number of hidden layers and neurons are changed according to each training results. In Figure 9a, Spark with a four-node cluster shows equal performance to GeForce GPU. Time taken to process the input images are taken in the y-axis, whereas the various platforms used for comparison are taken on the x-axis. The in-memory computation and Resilient Distributed Dataset (RDD). This provides a functional interface to partition the data across the cluster.

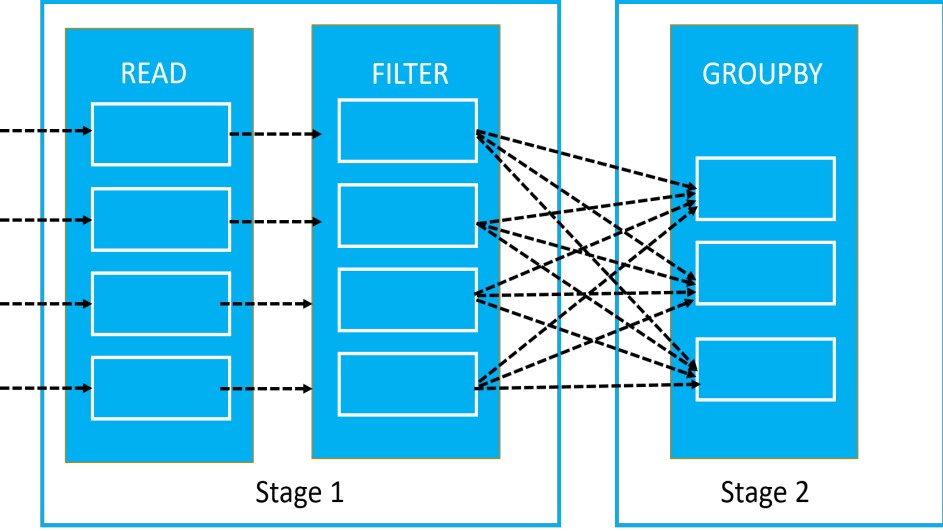

**Figure 7.** Example of a Spark shuffle.

We converted the Dicom scan image of the patients into RDDs for loading and passing through the nodes. We adopted HDFS as a file storage system for our lung nodule dataset and Spark for training the model. Even though a single node spark did not show promising results than a single GPU, clustered nodes can process the images in a few seconds. Overall, for our model and the training dataset, a GPU can process and train the images in 1678 s. Figure 9b shows the variation in batch size with all the model configurations. The throughput metrics are all in MB per second. For this investigation, we only considered the best results from each category.The main focus was to present the throughput difference between Spark and GPU and CPU TensorFlow. We found that model workload remains almost stable for Spark and GPU, yet Spark gives better throughput marginally in Figure 9c,d.

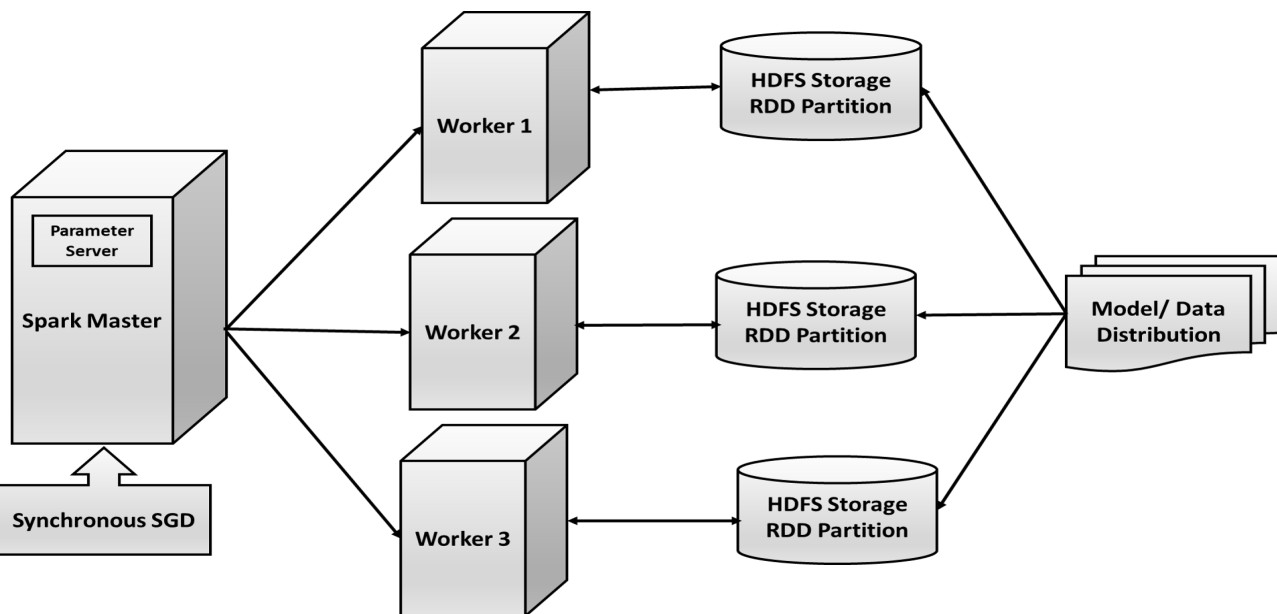

**Figure 8.** General architecture of Distributed Apache Spark and deep model.

When the dataset is tremendous and could not fit into single GPU memory, we can use distributed spark clusters to achieve convergence faster. TensorFlow on Spark provides fast processing in a distributed environment. The batch size of the data and the learning rate should be fixed according to the training accuracy to increase the throughput. In such a case, before a model gives an impressive performance, it needs to be trained repeatedly to achieve the proper hyper-parameters. Hence, Spark distributed training leads to better results when the deep neural network is complex and the dataset size is large. If we have shallow architecture and can fit the training data into a single machine, parallelism leads to computational overhead. We can adapt this setting to serverless architecture, as well. AWS, Microsoft Azure, and Google Cloud Platform are three gigantic heads in Cloud platforms. All three of them use Apache Spark on their cloud run for distributed parallel computing. Apache Spark performs well than GPU and has fault tolerance for a dataset. Spark has a replication system that allows it to reconstruct the data frames by itself in failure or crash. Spark also comes with built-in tools for machine learning algorithms and deep learning models, which lets us apply the ML model directly on batch data or real-time streaming data without any latency. It can be scaled horizontally and vertically without major hassle in coding and our experimentation results. Nevertheless, in the case of GPU, scaling is tedious, both concerning cost and coding.

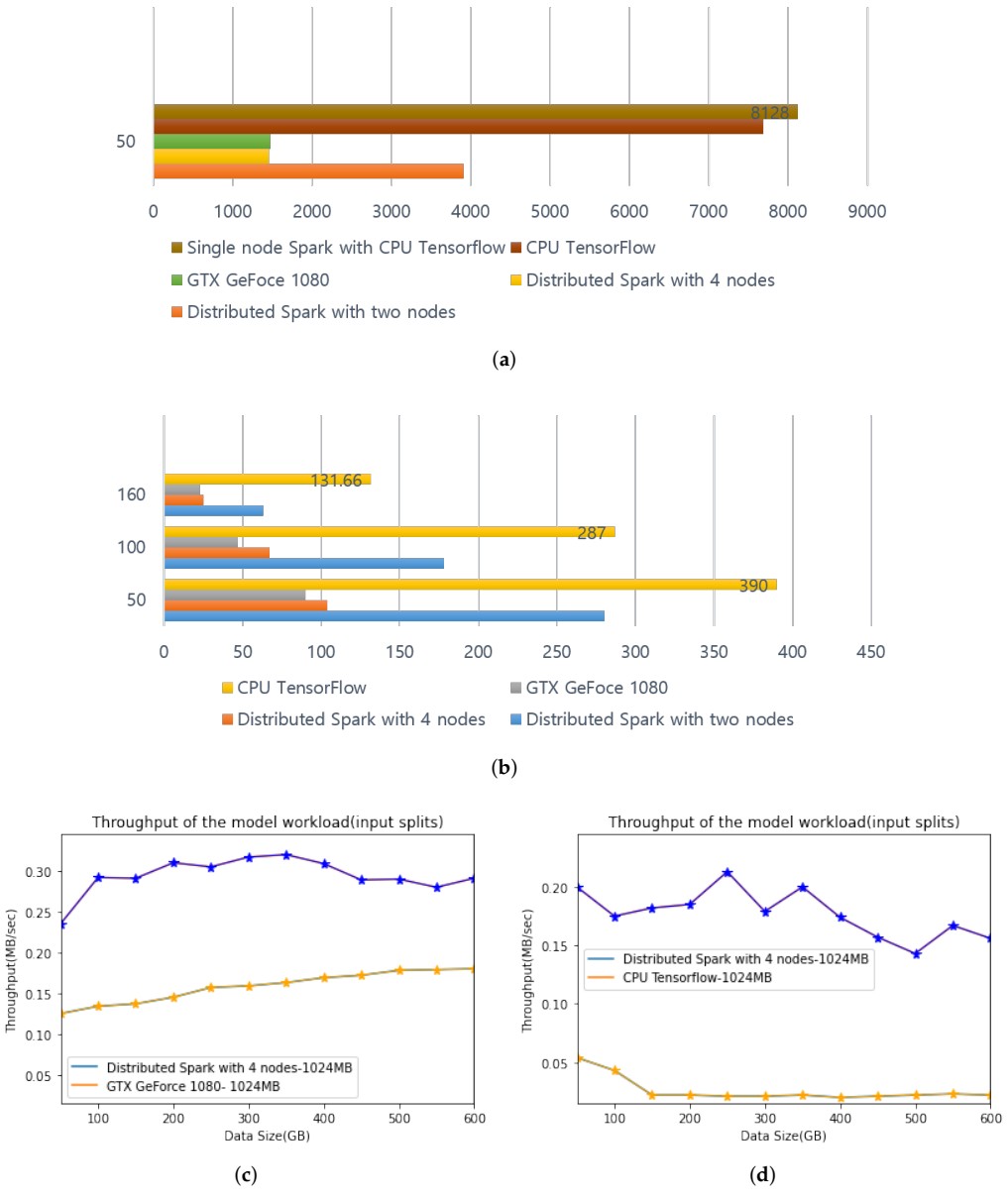

**Figure 9.** A set of four subfigures: (**a**) describes the Performance comparison between Apache Spark, GPU, and CPU; (**b**) describes the Performance comparison between Apache Spark, GPU, and CPU by varying the batch size; (**c**) describes the Throughput of model workload; (**d**) describes the Throughput comparison between spark and GPU.

## 5. Discussions & Future Work

This research intends to identify lung nodules using a deep model trained using the Apache Spark environment. Our paper enhanced the accuracy for detecting the lung nodules in patients scans by our proposed deep learning model NG-CNN. By eliminating the visual Gaussian noise in the CT scans, we can classify the lung scans' nodules at an early stage, which is notably crucial in predicting the disease to save a life. In addition to our proposed model, we also performed a performance evaluation to find the most reliable platform to run the deep learning model. Apache Spark shows promising results when integrated with distributed TensorFlow. As mentioned above, Spark has in-memory processing as it keeps intermediate results in RAM and built-in libraries for data analysis, machine learning, streaming live data, and graph analysis. Moreover, it also offers lazy evaluation and maintains a series of transformations with fault-tolerance. Hence, unless we perform the tasks, Spark can memorize the overall transformations. Spark has data locality,

better deal with failures and stragglers, and on top of everything, it is open-source. It's modern yet simple to use APIs to manage CPU, memory, and storage resources down to a granular level. Apache Spark is one of the most popular open source project in the Big Data landscape.

Our model is trained with the available open-source dataset. The accuracy can be improved further when the model is deep with many convolutional layers. The dataset needs to be enriched by using data augmentation, which will increase the dataset's size in our future work. Thus, the model will not over-fit the data with deep layers and also gives maximum accuracy. Our future work involves data augmentation for dataset increment and training the model in a deeper network.

**Author Contributions:** Conceptualization, methodology, software, validation, formal analysis, investigation, resources, data curation, writing—original draft preparation, writing, review and editing, N.J.V.; supervision, D.R.S.; project administration, C.S.N.; funding acquisition, C.S.N. All authors have read and agreed to the published version of the manuscript.

**Funding:** This research was supported by Basic Science Research Program through 631 the National Research Foundation of Korea (NRF) funded by the Ministry of Education (NRF 632 2019R1I1A1A01063278). And this work was supported by INHA UNIVERSITY Research Grant.

**Data Availability Statement:** The dataset we used in our research is publicly available dataset and can be downloaded from the following website, https://luna16.grand-challenge.org/Data/.

**Conflicts of Interest:** The authors declare no conflict of interest.

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
