# Peer review of "Nodule Detection with Convolutional Neural Network Using Apache Spark and GPU Frameworks"

_applsci, doi:10.3390/app11062838_

Round 1

Reviewer 1 Report

The paper presents a method to detect lung nodules by using convolutional neural network to provide a performance comparison based on different frameworks. Experimental data are presented and well described and they show that the open source Apache Spark framework is suitable for parallel training computation by offering high accuracy; however, the results are platform and hardware-dependent so they can change depending on different system architectures.
A better formatting is needed since some figures and captions are not centered with the text and are too small, i.e. Fig. 4, 5, 7, while Fig. 1 is too large based on its content. The same happens with some tables, i.e. Table 1,2,3.

Conclusions should be improved in order to better reflect the results.

Author Response

The authors are very much thankful to the reviewers for their precious time and helpful advice towards the improvement of the manuscript. Based on the advice of the reviewers following are the significant common changes. We appreciate the inputs and suggestions given by the reviewers. Their inputs definitely helped to improve our manuscript. Now through this letter, we would like to address each reviewer's comment one by one.  The reviewer's comments with their replies are as follows

Reviewer 2 Report

Authors present a convolutional neural network for the detection of lung nodules. The results seem pretty good compared to other methods shown in the paper.

But the writing of the paper is very confusing. The introduction is ok (although verbose, and the lack of paragraphs make very hard the reading), but the way the noise is introduced is verbose and confusing. The equations look awful with sigma and many other issues; the most surprising typo where the correntopy mutates to correntrophy (I guess authors use word so it not difficult to explain these issues and the awful look of the equations). Authors need a lot of words to explain correntopy and the explanation is not very rigorous as when the noise is introduced (they talk about Gaussian distribution, but nothing is said about the white spectral distribution)

Even the figures also look far from standard quality; fig. 1 is large; fig 9 is plenty of lines, ...

The organization of the paper is also inappropriate. The theory and results are mixed. For example, you must explain the CNN architecture in detail (how the noise is injected, and so on) in the methods and to show the performance in the results; the same for the hardware comparisons. 

Author Response

(The authors gave the same response as above.)

Reviewer 3 Report

  • Numerous language and editing issues, albeit minor, should be fixed (some examples: line 34 "classifies" should be "classify"; the sentence in line 43-45 has grammatical error, as does the sentence in lines 211-213; line 84 "Non- Gaussian" contains an extra space to be inconsistent with other occurrences of the same term; line 296 "Ct" should be "CT"; paragraphs starting at line 118, 179, 368, 465, and 520 are not indented as other paragraphs; the case of subsection titles 3.2 and 3.3 are inconsistent; Figure 9 is placed before Figure 8)
  • More serious concerns follow:  [1] in lines 314-316, it's stated that "Figure 1 depicts the architecture of our proposed model, defining the number of convolutional layers, ReLu layers, max-pooling, and fully connected layers." while no such information is found in Figure 1.  [2] Two lines after eq. (10), "qh is the bias ..." while neither eq. (9) nor eq. (10) contains qh[3] Two lines above eq. (13), it states "...the final graph of the training loss is given in figure 2.a.", Figure 2 (a) is a set of 4 images and it's unclear how do they show the training loss, explanation is needed.  [4] The left side of Eq. (13) has Jcost(θ); two lines below, it states that "J(cost) is the correntropy cost function", so what is Jcost(θ)?  [5] The headings of Table 1 and Table 2 are inconsistent, with Table 1 containing two columns of 'CNN"; also, is "DBN" the same as "DBF" in line 424?
  • The presentation could be more organized and the writing be made more concise, while more description and explanation should be provided where useful to help the reader; for example, immediately after eq. (1), the authors should explain what is Li, f and W, likewise, in eq. (7), the notation g(x)Δ deserves an explanation.

Author Response

The authors are very much thankful to the reviewers for their precious time and helpful advice towards improving the manuscript. Based on the direction of the reviewers, the following are the significant common changes. We appreciate the inputs and suggestions given by the reviewers. Their inputs helped to improve our manuscript. Now through this letter, we would like to address each reviewer's comment one by one.  The reviewer's comments with their replies are as follows

Round 2

Reviewer 2 Report

I appreciate the efforts from the authors to improve the quality of the paper in such a short time. But I still think that the writing of the paper is far from the standard quality for research papers. The size of the equations abd figures have been reduced but this is not the real problem.

I am even more surprised that you use a Latex template; this is the best in Latex, formatting. I do not understand how the equations still look awful, even they are out of the margins (eq.15), the multiplication symbol * instead of . and many more.

And the text contains some strange sentences: for example, after eq. 7, "the square of the Gaussian pdf sigma^2 is variance. In our model, sigma>0 takes only positive values and hence its called normalized kernel" first sentence makes little sense;  and of course variance is always positive, and what do you mean "normilized"? you mean your kernel is the normal (Gaussian)? because I do not see any normalization (divide by a factor so integral equals to 1 or any other thing you think "normalize" means)

Author Response

(The authors gave the same response as above.)

Reviewer 3 Report

The authors made corrections according to the first review.  A "spot check" of the revised manuscript, however, still found errors (of language) and raised many technical questions.

I will give one example of a technical issue:  in lines 332-334, "However, too many features will damper the classification accuracy. We chose 14 features, mainly based on geometric shapes and texture."  This is hardly convincing feature selection, as there is no telling that more (or perhaps less) features might lead to better results.   The authors are recommended to do more experiments with a large variety of feature sets, or provide full justification why the 14 (out 240) is the optimal feature set.

The writing, though somewhat improved from the inital draft, still leaves much to be desrired as there are numerous grammatical, punctuation, typing, and editing errors.    Just a few examples follow:

--lines 300-301 "To defend the robustness, we apply Jacobian norm mapping is analysed which is a non-linear mapping value of encoding function f."    The grammatical error in the sentence makes it incomprehensible.

--step 3 of Algorithm 1 "Noise is, identically distributed, and independent of any other signals in the system."  Both commas in the sentence can be removed.

--step 9 of Algorithm 1 starts with a comma.

--line 406  "The section produces results two-fold."  This poorly-written sentence can be understood only after reading the whole paragraph.

--line 594 "which is notable crucial in predicting the disease to save a life".   Either delete "notable" or say "notably crucial".

--line 599 "... builtin libraries" should be built-in.

--line 606-607, the sentence "Apache Spark is the most transformative open source project in the Big Data landscape." reeks of subjectivity, if not commercialism.   Why not just describe it as "one of the most transformative (or popular) ..."

Author Response

(The authors gave the same response as above.)

Round 3

Reviewer 2 Report

Thanks for your answers and effort to improve the paper